# Low Birth Weight, the Differentiating Risk Factor for Stunting among Preschool Children in India

**DOI:** 10.3390/ijerph19073751

**Published:** 2022-03-22

**Authors:** Shiva S. Halli, Rajeshwari A. Biradar, Jang Bahadur Prasad

**Affiliations:** 1Department of Community Health Sciences, Rady Faculty of Health Sciences, University of Manitoba, Winnipeg, MB R3E 0T6, Canada; 2Department of Epidemiology and Biostatistics, KLE University, Belgaum 590010, India; rajeshwaribiradariips@gmail.com (R.A.B.); jbiips12@gmail.com (J.B.P.)

**Keywords:** low birth weight, stunting, national family health survey, India

## Abstract

Background: The prevalence of low birth weight (LBW) is a major public health issue in India; however, the optimal growth pattern for such infants is not clear. The purpose of this study is to understand the causal association between LBW and stunting of preschool children in India. Methods: The National Family Health Survey-4 is a large cross-sectional survey based on a nationally representative sample of 699,686 women in the age group of 15–49 years and was conducted during 2015–2016 in India. The study used the children’s file with a sample of 259,002 of 0–59 months for investigation. Results: The data revealed that 38.7% of the children in India were stunted. The bivariate analysis revealed that, of the women who did not attend any antenatal care (ANC) meetings, 46.8% had stunted children compared to the women who attended more than three ANC meetings, which 30.7% had stunted children. The low birth weight children experienced a much higher chance of stunting compared to children with a normal birth weight (44.3% vs. 33.8%). The multivariable odds ratios of logistic regression, after adjusting for the confounding characteristics, showed that pregnant women attending more than three ANC meetings compared to not attending any ANC meetings experienced a 19% lower adjusted odds ratio (AOR) of having stunted children (AOR = 0.81; CI 0.78, 0.85; *p* < 0.001). Another important variable, such as women with underweight body mass index (BMI) compared with normal BMI, had 6% higher odds of having stunted children (AOR = 1.06; CI 1.03, 1.10; *p* < 0.001). Similarly, women who belong to the Scheduled Caste compared to the General Caste had 36% higher odds of having stunted children (AOR = 1.36; CI 1.30, 1.42; *p* < 0.001); and children aged 13–23 months compared to children up to one-year-old or younger had 141% higher odds of being stunted (AOR = 2.41; CI 2.32, 2.51; *p* < 0.001). The conspicuous finding is that LBW babies, after adjusting for other important confounding factors, such as BMI and ANC, experienced 19% higher odds of stunted children (AOR = 1.19; CI 1.14, 1.24; *p* < 0.001) compared to normal birth weight babies. Conclusions: The results revealed LBW is associated with stunting of preschool children in India.

## 1. Background

Stunting in early childhood is a major cause of various short-term and long-term consequences, particularly with an increase in childhood morbidity and mortality [1], delayed growth and motor development [2], and eventually educational and economic consequences [3]. To address this growing concern around the health needs of mother and child, the United Nations (UN) brought together the heads of states at the UN Headquarters to commit to the Millennium Development Goals (MDGs) in 2000 to achieve specific targets, with a deadline of 2015. Among the MDGs, Goal 4 is committed to reducing infant and child mortality. The United Nations became the catalyst to inspire people from around the world to take action in support of the MDGs. Taking the momentum forward in the post-2015 era, the Sustainable Development Goals (SDGs) aim to complete the work started by the MDGs and build on them in a more holistic manner [4]. For instance, SDG Goal 3 aims to “Ensure healthy lives and promote well-being for all at all ages” [5]. It includes a reduction in child deaths due to preventable causes. One of the thirteen specific targets for Goal 3 included, and to be achieved by 2030, was to end preventable deaths of children under five years of age, with all countries aiming to reduce under-five mortality to at least as low as 25 per 1000 live births [4]. India is moderately on track to reducing under-five mortality, but a key risk factor for child survival is stunting, mainly due to the undernutrition in expectant mothers [6]. In order to reduce the high social and economic cost of stunting, many countries have adopted the World Health Assembly target of achieving a 40% reduction in stunting by 2025. Meeting this target requires a significant reduction in stunting in India as it contributes to nearly one-third of the world’s total population of stunted preschoolers [7]. The challenge has been to understand the underlying determinants of stunting in India [8]. Many studies in different disciplines have indicated that the stunting problem in India is complex [9,10,11]. The possible explanations include economic growth and agricultural production [5,12,13], poor hygiene [5,12,13,14], discrimination against women and female children [15], anaemia before and during pregnancy [16], and very poor feeding practices among infants and children [11]. For targeting and planning purposes, it would be too difficult to prioritise amongst these multiple factors.

Low birth weight is also related to preterm delivery or intrauterine growth restriction, or both. It can be a predisposing factor to growth attainment after birth and important risk factors for spontaneous preterm delivery are short cervical length, raised cervical-vaginal fetal fibronectin concentration [17], and lifestyle factors, such as tobacco smoking [18]. There are also environmental factors, such as exposure to air pollution during pregnancy, especially in urban settings where it can be an important risk factor for preterm births [19]. This implies that if low birth weight is one of the risk factors for stunting among preschool children, any policy formation to address low birth weight ought to consider lifestyle as well as environmental factors, specifically smoking and air pollution.

The prevalence of low birth weight (LBW) is a major public health issue in India and is higher than the global prevalence. However, the optimal growth pattern for such infants is not clear, though it has been linked to an increased risk of later adiposity, insulin resistance, and cardiovascular disease [20]. Other studies also found that LBW was associated with sub-optimal cognitive development and the subdued growth of internal organs. This could affect cognitive ability later in life and increase the chances of chronic diseases [21]. In spite of LBW being the major public health problem, there are only a few studies conducted worldwide on its effect on stunting among children, including one in Africa that found that the growth of LBW babies is well behind the growth of normal weight babies [22], and another in Indonesia which found LBW as the most dominant predictor associated with stunting among children aged 12–23 months [23]. Though India has a higher prevalence of LBW globally, we have not come across any study directly linking LBW with the stunting of children below five years. Hence, an attempt is made in this study to analyse the impact of LBW on the likelihood of stunting among Indian children. By focusing on India, using the most recent and available national data set, this study provides the necessary empirical evidence that will contribute to the present theoretical debate, and emphasises the importance of focused strategic planning and evidence-based programming to achieve a 40% reduction in stunting by 2025.

## 2. Methods and Materials

### 2.1. Sampling Method

The study is based on the data from the children’s file of the National Family Health Survey-4, conducted during 2015–2016. The National Family Health Survey-4 (NFHS-4) is a large national survey that provides representative information for all the States and Union Territories of India. It is conducted by the International Institute for Population Sciences, Mumbai, and the detailed description of the sampling design is provided in the published NFHS-4 report [24]. The survey adopted a two-stage cluster sampling design. In the first stage, primary sampling units were selected to represent rural (villages) and urban (census enumeration blocks) areas of India using the 2011 Census as a sampling framework. A systematic random sampling design was used in the second stage to select households from selected rural and urban clusters, and from the selected households, study respondents were selected. The NFHS-4 provides data on 259,627 children aged 0–59 months. After excluding 11,884 dead children, 3235 children were missing information on their age, anthropometric measures—including 7372 children and flagged observations for 12,134 children—a final analytic sample of 225,002 children from NFHS-4 was used for the analysis (Figure 1). Sample size and data: Consisted of a household survey covering 640 districts from 29 States and 6 Union Territories of India. To cover the entire country, 28,586 primary sampling units (PSUs) were selected, out of which 28,522 clusters’ fieldwork was completed. A total of 601,509 households were interviewed using a structured questionnaire with a response rate of 98% with due respect to confidentiality. Outcomes: The outcome variable is stunting among children under five years. The NFHS collects information on children’s height, and the height measures were converted into age-specific Z-scores using World Health Organization (WHO) child growth standards. Based on WHO’s child growth standards, stunting of children was defined as height-for-age Z-score less than minus two standard deviations (-2 SD) from the median of the reference population [25].

### 2.2. Predictors

The main predictor of interest for the study is LBW. The definition used for LBW was a birth weight of less than 2500 g. The trained interviewers of the National Family Health Survey took the birth weight of the babies from written records. If the written records were not there, the information was collected from the baby’s mother. One of the important background characteristics in health research is Caste. According to traditional theory, Castes are originated from the body and the social hierarchy was assigned to organs of the body. From the head came the Brahmins, Kshatriyas from the arms, Vaishayas from the thighs, and Shudras from the feet. Hence, Brahmins were priests performing ceremonies, including preaching; Kshatriyas were meant to provide protection; Vaishayas were meant for farming or business; the Shudras are the lowest in the social hierarchy and were meant for serving all others. These four castes also have many sub-castes, and Shudras are considered Scheduled Castes in the study. In order to profile the stunted children, appropriate background characteristics were included. The selected characteristics include children’s age, rural-urban residence, broad categories of caste (Scheduled Caste/Scheduled Tribe (SC/ST, other backward castes (OBC) and Others)), religion (Hindu, Muslim, and Others), birth order, sex of the baby and immunisation status. In addition to this, the mother’s characteristics, such as her age at marriage, place of delivery, education level, number of antenatal care (ANC) visits, and breastfeeding practices were included. The LBW of a baby and body mass index of a woman, which was computed by the survey authorities using women’s weight and height, could be influenced by the nutritional status of a woman [16]. Hence, in the multivariable analysis, the interaction between LBW and body mass index (BMI) was included.

### 2.3. Statistical Analysis

Bivariate analysis was conducted to describe and understand the association of background characteristics with stunting children under five years using the model Chi-square. The backward stepwise multivariable logistic regression was used to assess the individual risk of the background characteristics after adjusting for the remaining characteristics, since many of the background characteristics could be highly associated with one another and may lead to a problem of multi-collinearity. In order to make sure that it was not a problem, the matrix of Pearson product-moment correlation was computed and found that none of the correlations were of concern for the purpose of including them in the multivariable logistic regression. The main variable of interest among the background characteristics is LBW. In order to assess its risk with stunting of children, important confounders, such as BMI, ANC, and the interaction terms between them were adjusted in addition to other background characteristics. To include the interaction terms, a separate model was performed. Moreover, the results were adjusted by applying appropriate statistical weights. The logistic regression adjusted estimates were presented in the form of adjusted odds ratios (AORs) along with a 95% confidence interval (CI). Moreover, to investigate the degree of sensitivity, sensitivity analysis was conducted for LBW using the receiver operating characteristic (ROC) curve (Figure 2). The analysis was conducted using IBM’s Statistical Package for the Social Sciences (SPSS), (version 20.0. Chicago, IL, USA).

## 3. Results

The bivariate association between stunting of children and their background characteristics is presented in Table 1. The table revealed that 38.7% of the children in India were stunted in 2015–2016. The results also indicated that women who marry before the age of 18 years were more likely to have stunted children (43.8%) compared to women who marry after the legal age of marriage (34.9%). Other significantly associated background characteristics with stunted children compared to their respective counterparts were rural residence (41.5% vs. 31.3%), Scheduled Caste (SC) (43.1%), and Scheduled Tribe (ST) (44.2) compared to general caste (31.0%) and LBW (44.3% vs. 33.8%), respectively. Similarly, as the birth order of a child increased, there was a systematic increase in the stunting of children. A strong inverse association was found between the education of reproductive women and the stunting of children. The antenatal care (ANC) visits by pregnant women, and institutional delivery received good support. For instance, the women who attended more than three ANC visits had 30.7% stunted children compared to women who did not attend any ANC experienced 46.8%. These are all bivariate associations, though statistically significant and hence, tentative until they are verified by multivariable analysis, adjusting for possible confounders. As indicated earlier, for the variable of interest, LBW, sensitivity analysis was conducted. The area under the ROC curve is 0.67 (95% CI: 0.66–0.67), indicating that LBW as a diagnostic tool for stunting among children is accurate at discriminating between those children who are stunted and those who are not.

The multivariable odds ratios of logistic regression, to understand the independent association of each of the background characteristics with stunting of children after adjusting for the remaining characteristics, are presented in Table 2. The table revealed that female children experienced 14% lower odds of stunting compared to male children (AOR = 0.86; CI 0.84, 0.88; *p* < 0.001). There is no significant difference in the likelihood of stunting of children among women delivering in public health facilities compared to home deliveries. However, pregnant women attending more than three ANCs compared to not attending any ANC have experienced 19% lower odds of having stunted children (AOR = 0.81; CI 0.78, 0.85; *p* < 0.001). Similarly, women with underweight BMI compared with normal BMI had 6% higher odds of having stunted children (AOR = 1.06; CI 1.03, 1.10; *p* < 0.001). The most conspicuous finding of interest is that LBW babies, after adjusting for other important confounding factors, such as the age of the child, education, BMI, and ANC, experienced 19% higher odds of being stunted (AOR = 1.19; CI 1.14, 1.24; *p* < 0.001) compared to normal birth weight babies. The other observed associations in the bivariate table between the age of children, education of reproductive women, a social hierarchy based on caste and religion with stunting of children were supported by multivariable results of Table 2. For instance, women with more than secondary education compared to no education experienced 51% lower odds of having stunted children (AOR = 0.49; CI 0.47, 0.52; *p* < 0.001). Similarly, women who belong to Scheduled Caste compared to General Caste had 36% higher odds of having stunted children (AOR = 1.36; CI 1.30, 1.42; *p* < 0.001); and children aged 13–23 months compared to children up to one-year-old had 141% higher odds of being stunted (AOR = 2.41; CI 2.32, 2.51; *p* < 0.001). Regarding religion, Muslims had 9% higher odds of having stunted children compared to Hindus (AOR = 1.09; CI 1.05, 1.14; *p* < 0.001).

## 4. Discussion

According to National Family Health Survey-4, the stunting of children below the age of five years is as high as 39% in India [24]. Based on a recent study in Indonesia, the most important determinant of stunting of children was low birth weight [23]. As mentioned earlier, surprisingly, there is no study in India to verify whether LBW is associated with the stunting of children. However, there are studies to show that the prevalence of underweight children was significantly higher among underweight women compared to their normal BMI counterparts [26] and those with no antenatal care [27]. As expected, women with underweight BMI experienced a 6% higher odds compared to women with normal BMI, and women attending more than three-plus ANCs experienced 19% lower odds compared to women attending no ANC. Surprisingly, even after controlling for education and residence, the socio-economically marginalised and stigmatised groups, such as Scheduled Caste and Scheduled Tribe, tend to experience significantly higher odds (36% and 32%) of stunted children compared to the general caste. A recent study conducted in South Asia has also shown that socio-economically disadvantaged groups experienced higher stunting of children [28]. Because of social isolation and marginalisation, Scheduled Caste and Scheduled Tribe groups are deprived of visits with Accredited Social Health Activists (ASHAs, frontline health workers). They are expected to contact and counsel women on antenatal care and the importance of institutional delivery and postnatal care [29,30]. The lack of ASHAs’ home visits appeared to have influenced many women by the lower utilisation of antenatal health care services among historically disadvantaged social groups, such as Scheduled Caste and Scheduled Tribe (SC/ST).

Moreover, this could result in poor communication between SC/ST communities and other upper caste communities, leading to a lack of information regarding health care services and utilisation. For instance, hierarchies based on local constellations of a set of identities—caste and religion—can be understood as having been shaped through the interplay of social, political, and economic processes that have influenced groups’ relative power and resources over time. These cultural and religious dimensions of identity should be seen as interacting with political and economic processes to shape socioeconomic inequities over time, resulting in poor maternal health outcomes and, in turn, poor child outcomes [31,32]. Indeed, the religious minority group in this study, namely the Muslim community, also experienced higher stunting of children compared to the Hindu majority group. In order to strengthen ASHAs’ efforts to encourage women to have ANC check-ups and stronger facility-based services, community platforms may be needed, especially for disadvantaged communities of SC/ST [30] and Muslim women.

The multivariable regression results of the study also indicated the education of women as one of the important factors in stunting of children, meaning lower education is associated with higher stunting. Studies conducted in many parts of the country have shown that a frontline worker’s visit is greatly associated with the utilisation of maternal health care services for women with lower education levels [33,34,35]. Others have argued that overall rates of maternal health care services’ utilisation in India have increased significantly, mainly due to Janani Suraksha Yojana (JSY) incentives. The frontline workers’ role in promoting these incentive services was important for women with little or no education [36]. Women with higher education may not need information or motivation to access free government health care service schemes or assess the quality of care provided in public health facilities. They may utilise good quality private health care services if necessary, because of their resources. A surprising result was that female children were 13% less likely to experience stunting compared to male children. Other studies have also shown that more boys were severely stunted compared to girls [37]. A study conducted in Bhubaneswar observed that more boys (63.3%) were stunted than their counterparts (54.7%) [38].

The variable of interest in this study is LBW. It turns out that LBW is associated with stunting of preschool children in the multivariable logistic regression after controlling for important confounders, such as children’s age, mother’s education, BMI, and ANC. A recent study in India on maternal nutritional status found women suffering from anaemia are at higher risk of poor birth outcomes, such as preterm birth and low birth weight due to weak intrauterine growth [26]. The studies have documented that half of the expectant mothers, children, and adolescent girls in India suffer from anaemia [39]. At least half of the burden of anaemia is assumed to be due to iron deficiency, and both folic acid and iron deficiency during pregnancy are important factors for preterm delivery, anaemia, low birth weight, and in turn, increased stunting among children [4,5]. The World Health Organization suggested that those pregnant women who attend ANC meetings should be given a recommended dose of 30–60 mg iron and 400 mg folic acid [40]. In India, iron, and folic acid (IFA) consumption has been low among pregnant women despite IFA supplements being distributed for free, especially during ANC visits. This may partly be due to full (at least four or more visits) ANC coverage nationally being only 59% during their last pregnancy [41]. The policymakers also believe that there is a problem with adherence. This is not entirely true, as there seems to be a fundamental health system problem. According to one study, there is a serious problem of stock and untimely supply of IFA supplements in government facilities, rather than poor adherence to taking the supplement by women, which has been a major concern for policymakers [42]. Among those women who attended one or two ANCs, less than 50% of the pregnant women received the recommended 100 or more IFA supplements [42].

In addition to maternal malnutrition, infant, and child undernutrition is also an important risk factor of stunting among children. In order to understand feeding practices among preschool children in India, the corresponding variables in the data set were the early initiation of breastfeeding, exclusive breastfeeding, a timely introduction of complementary foods, and an adequate diet. Unfortunately, data on these variables is not complete. For instance, the survey covers 640 districts of India, but data on exclusive breastfeeding are only available for 425 districts, and data for the timely introduction of foods are only available in 186 districts. Perhaps because of the incompleteness of data on breastfeeding, the counterintuitive result of children who were breastfed <6 months were 16% less likely to be stunted, was observed in Table 2. For instance, the district level data analysis has shown that only about 40% of children were breastfed within an hour and about 55% of children were exclusively breastfed [11]. The more serious problem was complementary feeding: less than 10% of children were given an adequately diverse diet [10]. Here is an opportunity for the Government of India to address child stunting by making sure that socially marginalised and economically backward groups do receive both solid and semi-solid complimentary food through frontline workers, such as ASHAs.

Some researchers argue that children of shorter gestational duration tend to be stunted, however, later, most such children catch up with their growth; however, about 10% of those children will remain stunted. When we analysed the data, we found that there are few cases with shorter gestational pregnancies in the sample. For instance, for 5 months’ gestational duration, there are only 24 cases and 16 of these were with stunted children (66.7%). Similarly, for 6 months gestational duration, there are 62 cases and of these, 34 were with stunted children (55%). Of those women with 8 and 9 months gestational durations, about 40% of those children were stunted, similar to the national average of 39%.

## 5. Conclusions

The main risk factors observed in the study for stunting of children in India are the age of children, education of women, and disadvantaged caste groups. These are expected risk factors highlighted in the literature. However, even after controlling for important confounders, LBW contributes to stunting of children in nearly one in five children. If the government of India is serious about the World Health Assembly target of achieving a 40% reduction in stunting by 2025, the main risk factors causing stunting of children in India, such as SC/ST groups and LBW need to be included in its policy formulation. For instance, ANC attendance seems to matter in the reduction in stunting, and hence, an increase in coverage of ANC, especially for socially disadvantaged communities, such as Scheduled Caste and Scheduled Tribe, is important. To increase the coverage among the SC/ST communities, it is important to increase the home visits of the Accredited Social Health Activists (ASHAs) to contact and counsel women on antenatal care and increase attendance in ANC meetings. To support ASHAs’ efforts to encourage SC/ST women to attend ANC meetings, stronger community platforms may be needed because of geographical isolation and eradicating “untouchability” and discrimination against these communities. It may also be necessary to sensitise frontline workers and other health care providers to address the stigmas and discrimination against SC/ST communities. Similarly, innovative interventions should be considered to address the problem of LBW. For instance, the Indian government has recently launched the National Nutrition Mission, or POSHAN Abhiyaan, with the objective to reduce malnutrition and to emphasize a diversified diet for children, and pregnant and lactating mothers. However, it is important that these schemes are effectively implemented. For instance, as highlighted in the discussion about the distribution of IFA supplements through ANC attendance to address anaemia among pregnant women, there is a serious problem of stock and untimely supply of IFA supplements in government facilities. The government should consider making special arrangements, including home delivery of the recommended 100 IFA supplements to disadvantaged groups, such as rural SC/ST members. Similarly, the government should also ensure that SC/ST groups receive both solid and semi-solid complimentary food through their community health workers, such as ASHAs and Anganwadi Workers.

## Figures and Tables

**Figure 1 ijerph-19-03751-f001:**
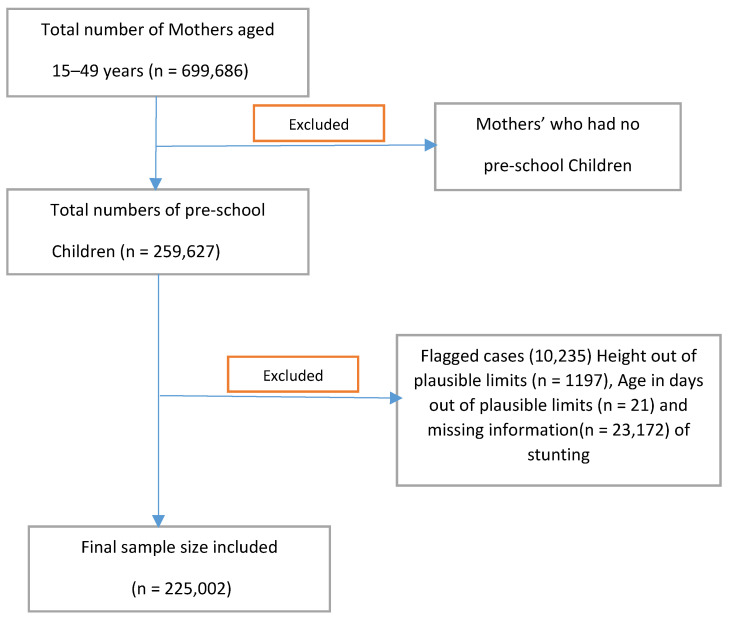
Flow diagram showing exclusions and final sample sizes of the study population, National Family Health Survey 2015–2016.

**Figure 2 ijerph-19-03751-f002:**
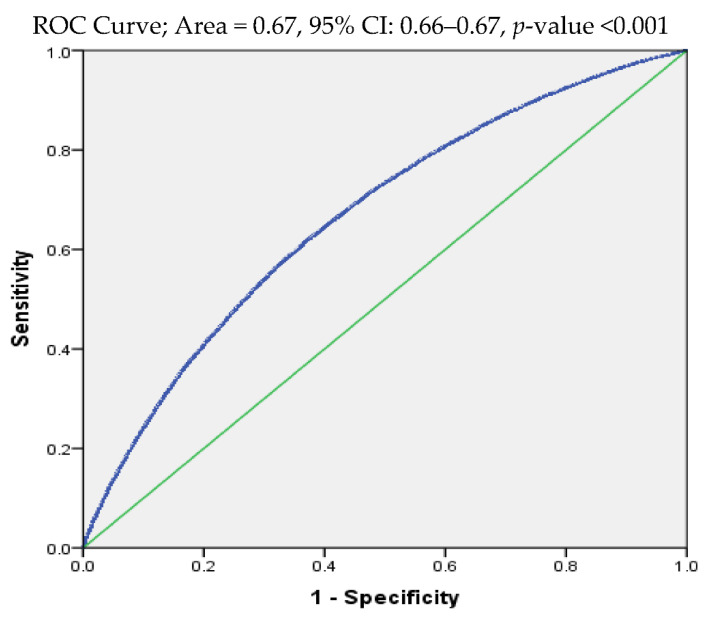
Receiving operative characteristic curve highlighting the results (area under the curve: 0.67 and 95%, CI: 0.66–0.67).

**Table 1 ijerph-19-03751-t001:** Prevalence (%) of stunting among Indian children by background characteristics.

		Stunted		*p*-Value *	Total Cases
		No	Yes		
Age at marriage	18 & above	65.1	34.9	<0.001	135,902
	<18 years	56.2	43.8		86,809
Place of residence	Urban	68.7	31.3	<0.001	53,483
	Rural	58.5	41.5		171,519
Religion	Hindu	61.2	38.8	<0.001	163,089
	Muslim	59.9	40.1		35,241
	Others	68.0	32.0		26,672
Caste	General	69.0	31.0	<0.001	39,399
	Schedule caste	56.9	43.1		42,540
	Schedule tribe	55.8	44.2		44,440
	Other backward caste	61.0	39.0		88,803
ANC visits	0	53.2	46.8	<0.001	28,461
	1–3	59.8	40.2		58,464
	3+	69.3	30.7		81,044
Place of delivery	Home	51.4	48.6	<0.001	54,095
	Public Facility	60.5	39.5		123,066
	Private facility	70.6	29.4		47,841
LBW	2500 & above grams	66.2	33.8	<0.001	141,573
	<2500 g	55.7	44.3		28,746
Currently on breastfeeding	No	62.7	37.3	<0.001	95,980
	Yes	60.3	39.7		119,142
BMI of mother	Normal	61.6	38.4	<0.001	139,618
	Underweight	53.9	46.1		53,285
	Overweight/Obese	72.6	27.4		31,530
Birth order	1	66.5	33.5	<0.001	83,046
	2	62.6	37.4		69,784
	3	56.9	43.1		36,228
	4 & above	49.3	50.7		35,944
Age of child (in months)	≥12	77.5	22.5	<0.001	45,686
	13–23	55.7	44.3		41,492
	24–35	57.0	43.0		45,096
	36–47	56.4	43.6		47,410
	48–59	59.7	40.3		45,318
Sex of child	Male	60.8	39.2	<0.001	116,360
	Female	61.9	38.1		108,642
Fully immunised	No	60.4	39.6	<0.001	94,897
	Yes	62.0	38.0		130,105
Highest educational level	No education	48.9	51.1	<0.001	68,978
	Primary	56.2	43.8		32,835
	Secondary	66.9	33.1		102,191
	Higher	78.9	21.1		20,998
Duration of breastfeeding	>6 Months	60.6	39.4	<0.001	125,721
	≤6 Months	70.6	29.4		47,520
	India	61.3	38.7		225,002

Note: * Determined by χ2 test; *p* < 0.05 considered significant.

**Table 2 ijerph-19-03751-t002:** Adjusted odds ratios (AORs) for the association of covariates with stunting of children in India.

		*p*-Value	AOR	95% CI for EXP(B)	
				Lower	Upper
Age at marriage	18 & above		1.00 (ref)		
	<18 years	<0.001	1.06	1.03	1.09
Place of residence	Urban		1.00 (ref)		
	Rural	<0.001	1.08	1.05	1.12
Religion	Hindu		1.00 (ref)		
	Muslim	<0.001	1.09	1.05	1.14
	Others	<0.001	0.81	0.78	0.85
Caste	General		1.00 (ref)		
	Schedule caste	<0.001	1.36	1.30	1.42
	Schedule tribe	<0.001	1.32	1.27	1.39
	Other backward caste	<0.0001	1.21	1.18	1.26
ANC visits	0		1.00 (ref)		
	1–3	<0.001	0.91	0.88	0.96
	3+	<0.001	0.811	0.777	0.846
Place of delivery	Home		1.00 (ref)		
	Public Facility	0.190	0.97	0.92	1.02
	Private facility	<0.001	0.85	0.81	0.90
LBW	2500 & above grams		1.00 (ref)		
	<2500 g	<0.001	1.19	1.14	1.24
BMI of mother	Normal		1.00 (ref)		
	Underweight	<0.001	1.06	1.03	1.10
	Overweight/Obese	0.594	1.01	0.97	1.069
Birth order	1		1.00 (ref)		
	2	<0.001	1.08	1.05	1.11
	3	<0.001	1.17	1.13	1.22
	4 & above	<0.001	1.32	1.26	1.38
Age of child (in months)	≥12		1.00 (ref)		
	13–23	<0.001	2.41	2.32	2.51
	24–35	<0.001	2.24	2.15	2.33
	36–47	<0.001	2.28	2.18	2.38
	48–59	<0.001	1.87	1.79	1.97
Sex of child	Male		1.00 (ref)		
	Female	<0.001	0.86	0.84	0.88
Highest educational level	No education		1.00 (ref)		
	Primary	<0.001	0.87	0.83	0.91
	Secondary	<0.001	0.67	0.65	0.69
	Higher	<0.001	0.49	0.47	0.52
Duration of breastfeeding	>6 Months		1.00 (ref)		
	≤6 Months	<0.001	0.84	0.81	0.87
Child Birthweight * BMI of mothers		<0.001	0.98	0.99	0.99

ref: Reference category; *: Interaction variable.

## Data Availability

All the data that is used in this paper has been archived in the Demographic and Health Surveys (DHS) public repository, where the data is accessible using the link: https://www.dhsprogram.com/data/available-datasets.cfm (accessed on 17 March 2022).

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
