# Peer review of "Low Birth Weight, the Differentiating Risk Factor for Stunting among Preschool Children in India"

_ijerph, 2022, doi:10.3390/ijerph19073751_

Round 1

Reviewer 1 Report

The manuscript has been revised well. I think this manuscript will be acceptable.

Author Response

Reviewer 1:

Comment: The manuscript has been revised well. I think this manuscript will be acceptable.

Response: We thank the reviewer.

Reviewer 2 Report

The main aim of the study was to understand the causal relationship associated with low birth weight and stunting of preschool children. The manuscript provides evidence of the problem as well as possible solutions that could be undertaken. The limitations of the study should be more defined, and the conclusion could entertain more solutions to the problem. Ones that would be easily attainable in the various caste settings. 

The article is complete and the topic i s relevant and presented in a well structured manner.. References are appropriate, there is not an overabundance of self-citations. The experimental design is sound and the results reproducible. Figures/tables/images/schemes are and appropriate properly show the data. They easy to interpret and understand and the data interpreted appropriately and consistently throughout the manuscript. 

SPecific details to address:

Page 2 check the grammar, thee is a citation missing and LBW should de defined here or in the methods section. Tie the two paragraphs together on page two as well.

Are SC and ST the same?

I believe the term is multivariable logistic regression.

Page 8: CHeck sentence structure in the newly added section.

Author Response

Reviewer 2:

Comment: The article is complete and the topic i s relevant and presented in a well-structured manner The limitations of the study should be more defined and the conclusion could entertain more solutions to the problem. Ones that would be easily attainable in the various caste settings.

Response: Based on the comments of the reviewer, the limitations of the study presented in the last two paragraphs of the Discussion section were revised. We also added a last sentence in the Conclusion section to indicate a possible solution to address problem of disadvantaged Caste groups.

Specific Comment 1: Page 2 check the grammar, thee is a citation missing and LBW should be defined here or in the methods section.

Response: We have checked the grammar and added the citation (Reference # 13). We have also defined LBW in the Methods section.

Specific Comment 2: Tie the two paragraphs together on page two as well.

Response: Done.

Specific Comment 3: Are SC and ST the same?

Response: No.

Specific Comment 4: I believe the term is multivariable logistic regression.

Response: Yes, and made sure that the term is used accordingly.

Specific Comment 5:  Page 8: CHeck sentence structure in the newly added section.

Response: Done

Reviewer 3 Report

Low Birth Weight, the Differentiating Risk Factor for Stunting among Preschool Children in India

This is an enormous study of a very important topic, stunting in India.  The available variables make this an impressive report.  The authors demonstrate the multifaceted and complex nature of this problem.  While the LBW-stunting association was their primary finding, there were so many other interesting and important results.  I have made several minor suggestions below.  I suggest a careful reading by an English reader would be helpful.

Abstract

About 1/3 of the way down the abstract “compared to of the women” should be “compared to the women”.

About 2/3 of the way down the abstract “AOR=0.1.06;” should be “AOR=1.06;”

Three lines down from there, “younger had 241% higher odds of stunted children (AOR=2.41;” should be “younger had 141% higher odds of stunted children (AOR=2.41;”.  The logic here is that in the line above the authors used “36% higher” when the AOR was 1.36.

  1. Background

In the Background and elsewhere, the authors mix the use of abbreviations and the full text.  For example, after MDGs is identified as the abbreviation for Millennium Development Goals on line 6 of the Background, the full phrase is used again on line 9.  Low birth weight is abbreviated LBW at the beginning of the second paragraph, but low birth weight is used again five times in the same paragraph.

  1. Methods and materials

2.1 Sampling method

The following section just before “Outcomes” is repetitive and can largely be removed. “The NFHS-4 provides data on 259,627 children aged 0-59 months. After applying exclusion criteria mentioned in Figure A1, a sample of 225,002 children of 0-59 months was included in the analysis. Details about methodology, along with complete information on survey design and data collection methods, were published in the NFHS survey report [21].”

2.3. Statistical analysis

A more complete description of the included confounders used in the Table 2 analysis would be helpful.  With so many variables analyzed, the reader wonders if different confounders were used in each analysis.

  1. Results

The sentence “The important variable of interest, low birth weight children, experienced a much higher chance of stunting than children with normal birth weight (44.3% v/s 33.8%).” is repetitive from 6 lines above.

Page 6, line 10: “AOR=0.1.06” should be “AOR=1.06”

Page 7: line 8: “one-year-old had 241% higher odds” should be “one-year-old had 141% higher odds”

Table 2: Please keep the significant figures to two to the right of the decimal place for the AORs.

Discussion

Would the authors care to comment on the interesting finding that children who were breastfed < 6 months were 16% less likely to be stunted? Counterintuitive.

Author Response

Reviewer 3:

Comment 1: This is an enormous study of a very important topic, stunting in India.  The available variables make this an impressive report.  The authors demonstrate the multifaceted and complex nature of this problem.  While the LBW-stunting association was their primary finding, there were so many other interesting and important results.  I have made several minor suggestions below.  I suggest a careful reading by an English reader would be helpful.

Response: We thank the reviewer for liking the paper. The paper is read by an English reader.

Abstract

Specific Comment 1: About 1/3 of the way down the abstract “compared to of the women” should be “compared to the women”.

Response: We thank the reviewer for noticing this and corrected accordingly. 

Specific Comment 2: About 2/3 of the way down the abstract “AOR=0.1.06;” should be “AOR=1.06;”

Response: Thank you and corrected it.

Specific Comment 3: Three lines down from there, “younger had 241% higher odds of stunted children (AOR=2.41;” should be “younger had 141% higher odds of stunted children (AOR=2.41;”.  The logic here is that in the line above the authors used “36% higher” when the AOR was 1.36.

Response: We agree with the reviewer and changed accordingly.

  1. Background

Specific Comment 4: In the Background and elsewhere, the authors mix the use of abbreviations and the full text.  For example, after MDGs is identified as the abbreviation for Millennium Development Goals on line 6 of the Background, the full phrase is used again on line 9.  Low birth weight is abbreviated LBW at the beginning of the second paragraph, but low birth weight is used again five times in the same paragraph.

Response: Thank you and changed in the text as suggested. 

  1. Methods and materials

2.1 Sampling method

Specific Comment 5: The following section just before “Outcomes” is repetitive and can largely be removed. “The NFHS-4 provides data on 259,627 children aged 0-59 months. After applying exclusion criteria mentioned in Figure A1, a sample of 225,002 children of 0-59 months was included in the analysis. Details about methodology, along with complete information on survey design and data collection methods, were published in the NFHS survey report [21].”

Response: Thank you and deleted the repetition.

2.3. Statistical analysis

Specific Comment 6: A more complete description of the included confounders used in the Table 2 analysis would be helpful. 

Response: In the section on Predictors, the description of confounders is included.

Specific Comment 7:  With so many variables analyzed, the reader wonders if different confounders were used in each analysis.

Response: The confounding variables are selected based on the review of literature and when many confounders used in the analysis there could be a problem of multi-collinearity and this has been checked. Moreover, we have used the backward stepwise multivariable logistic regression to assess the individual risk of the background characteristics after adjusting for the remaining characteristics.

  1. Results

 Specific Comment 8:  The sentence “The important variable of interest, low birth weight children, experienced a much higher chance of stunting than children with normal birth weight (44.3% v/s 33.8%).” is repetitive from 6 lines above.

Response: Thank you and deleted the repetitive lines.

Specific Comment 9:  Page 6, line 10: “AOR=0.1.06” should be “AOR=1.06”

Response: Thank you and the correction is made.

Specific Comment 10:  Page 7: line 8: “one-year-old had 241% higher odds” should be “one-year-old had 141% higher odds”

Response: Thank you and the correction is made.

Specific Comment 11:  Table 2: Please keep the significant figures to two to the right of the decimal place for the AORs.

Response: The AORs are restricted to two to the right of the decimal place. Regarding the p-values, presenting <.001 to <.00 can be misleading and hence, retained as they are. 

Discussion

Specific Comment 12:  Would the authors care to comment on the interesting finding that children who were breastfed < 6 months were 16% less likely to be stunted? Counterintuitive.

Response: Thank you and commented on the counterintuitive result. 

Reviewer 4 Report

This is my review on the manuscript entitled “Low Birth Weight, the Differentiating Risk Factor for Stunting among Preschool Children in India”. This is a major public health issue that authors are trying to investigate in relation to the stunting of preschool children.

It is important to cover in introduction implications for deeper causes regarding the low birth weight. LBW is surely related to preterm delivery and preterm delivery is related to several factors, with air pollution to be one of the most important among the environmental ones. Below there are some references that should be mentioned to cover that part:

https://www.thelancet.com/journals/lancet/article/PIIS0140-6736(08)60074-4/fulltext

https://pubmed.ncbi.nlm.nih.gov/25394641/

https://pubmed.ncbi.nlm.nih.gov/32563251/

https://pubmed.ncbi.nlm.nih.gov/33364740/

https://pubmed.ncbi.nlm.nih.gov/33921784/

Other than that, this is a very interesting study that provides significant socio-demographic aspects of the stunting.

Author Response

Reviewer 4:

Comment: It is important to cover in introduction implications for deeper causes regarding the low birth weight. LBW is surely related to preterm delivery and preterm delivery is related to several factors, with air pollution to be one of the most important among the environmental ones. Below there are some references that should be mentioned to cover that part:

Response: We thank the reviewer for the suggestion. We included in the introduction factors affecting LBW such as smoking and air pollution (e.g., references 17, 18 and 19 are added). We have also mentioned that any comprehensive policy formation to address stunting among pre-school children should not only address LBW and also the factors affecting LBW such as smoking and air pollution.

This manuscript is a resubmission of an earlier submission. The following is a list of the peer review reports and author responses from that submission.

Round 1

Reviewer 1 Report

This paper uses Indian NFHS 2015-2016 data to look at correlates of stunting among under-5 children. It does not focus on low birth weight (LBW), despite its title.

I have two main comments:

First, the authors do NOT identify “causal” association between LBW and stunting. There is nothing causal about a logit regression.

Second, and more importantly, this article was clearly written about the association between casts and religious minorities and stunting of under-5 children. It seems that the authors took that original article, included a few sentences about LBW and are now selling it as a new article that looks at the association between LBW and stunting. I find this practice extremely unscientific.

Author Response

Reviewer 1:

I have two main comments:

Comment 1: First, the authors do NOT identify “causal” association between LBW and stunting. There is nothing causal about a logit regression.

Response: The title of the paper is: Low birth weight, the differentiating risk factor for stunting among preschool children in India. To justify the title, we had to examine whether LBW is a significant risk factor for stunting. The outcome, Stunting, is a binary variable and hence, we performed logistic regression analysis in order to evaluate the probability that a child with a particular covariate pattern will have the stunting or not. Then used this probability to assign the child to an outcome group that reflects the child’s risk of stunting. The odds ratio for LBW   even after adjusting for all background characteristics is significant and increases the risk of stunting by 19% among LBW children compared to non LBW children. Furthermore, to make sure that the analysis is robust, Sensitivity analysis is conducted and ROC Curve is included (Figure A1). We have also conducted the backward stepwise procedure for multivariate analysis and the variable LBW was retained and turned out to be a significant risk factor.

Comment 2: Second, and more importantly, this article was clearly written about the association between casts and religious minorities and stunting of under-5 children. It seems that the authors took that original article, included a few sentences about LBW and are now selling it as a new article that looks at the association between LBW and stunting. I find this practice extremely unscientific.

Response: We are not clear about the comment.  Does the reviewer think that we had published an article on castes and religious minorities and stunting, and later “included a few sentences on LBW and selling it as new article”? We want to make it clear that we have not published any other article on “the association between casts and religious minorities and stunting of under-5 children”.

Reviewer 2 Report

My one question in the abstract is why the choice of 0-59 months?

In order to have better understanding for all readers I would define the term stunting. Not all countries use this term. Define the "caste" this will allow for better understanding across all cultures.

Author Response

Reviewer 2:

Comment 1: My one question in the abstract is why the choice of 0-59 months?

Response: This is a legitimate question because the recent literature including WHO advocates strategies for improving nutritional intake during the first 1,000-days (Schwarzenberg, S.J.; Georgieff, M.K. Advocacy for improving nutrition in the first 1000 days to support childhood development and adult health. Pediatrics 2018, 141, e20173716.).  However, based on extensive analysis of two successive National Family Health Surveys conducted in 2006 and 2016 in India, one recent study has shown that “child stunting was significantly concentrated among children entering preschool age (24 or above months). Further, the temporal reduction in stunting was relatively higher among children aged 36–47 months compared to

younger groups (below 12 and 12–23 months). Greater socioeconomic inequalities persisted in stunting

among children from 24 months or above age-groups, and these inequalities have increased over time” (Rajpal S, Kim R, Joe W, Subramanian SV. Stunting among Preschool Children in India: Temporal Analysis of Age-Specific Wealth Inequalities. International Journal of Environmental Research and Public Health. 2020 Jan;17(13):4702). In spite of Government of India’s concentrated efforts to address undernutrition in the first two years of birth has not had much impact. Moreover, the slow progress in the first two years has had adverse impact on later years. Hence, we have made the choice of 0-59 months and this is also consistent with other recent studies conducted in India.

Comment 2: In order to have better understanding for all readers I would define the term stunting. Not all countries use this term. Define the "caste" this will allow for better understanding across all cultures.

Response: We agree with the reviewer. Stunting is defined in the sampling method and data sub-section towards the end, the last sentence.  We have also provided the WHO reference for this. Similarly, in the same section caste is also described.

Reviewer 3 Report

The National Family Health Survey-4 (NFHS-4) is a large national survey in India. The author conclude that LBW is the significant risk factor for stunting of preschool children in India. Although this paper added some clinical data of the association between LBW and stunting of preschool children in India, there are few new findings.

Major comments:

1)  The major concern about this study is that LBW is the significant risk factor for stunting of preschool children and dose not add a novel finding.

2)

(1. Background) Many studies of different disciplines have indicated that the stunting problem in India is com-plex [8-10]. ...... very poor feeding practices among infants and children [10].

The aurthors shoud describe feeding practices of the children in this study. Very poor feeding practices are risk factor for stunting of prechool children in India, wheter they were LBW or not.

3) The aurthors should describe small for gestational age (SGA) children in this study.  After birth most SGA infants good catch-up growth and normalaize their height and wight. About 10 % of them continue to remain short  (<-2.0SD) and stunting (<-2.0 SD ).

Minor comments:

1)The authors should shorten background. The description about MDGs is too long.

2)

(2. Methods and materials  2.2. Predictors)

Hence, in the multivariate analysis, the interaction between low birth weight and body mass index is included.

(2. Methods and materials  2.3. Statistical analysis)

In order to assess its risk with stunting of children, for important confounders such as BMI, ANC and the interaction terms between them were adjusted in addition to other background characteristics.

Is “BMI of the mothers “ correct?

Author Response

Reviewer 3:

Comment 1: The major concern about this study is that LBW is the significant risk factor for stunting of preschool children and does not add a novel finding.

Response: Although stunting among children as well as LBW are studied extensively, as indicated in the background of the paper, there are only two studies conducted worldwide in linking the two, one in Asia and another in Africa. In spite of India has higher prevalence of low birth weight and stunting globally, we have not come across any study directly linking low birth weight with stunting of children below five years in India.  

Comment 2: (1. Background) Many studies of different disciplines have indicated that the stunting problem in India is com-plex [8-10]. ...... very poor feeding practices among infants and children [10].

The aurthors shoud describe feeding practices of the children in this study. Very poor feeding practices are risk factor for stunting of prechool children in India, wheter they were LBW or not.

Response: We agree with the reviewer that it is important to describe feeding practices of the children. We did think about including feeding practices but the data is incomplete on this. For instance, the survey covers 640 districts of India but data for exclusive breast feeding are available only for 425 districts and data for timely introduction of foods are available only for 186 districts. However, we have added a paragraph towards end of the discussion section on this. 

Comment 3:  The aurthors should describe small for gestational age (SGA) children in this study.  After birth most SGA infants good catch-up growth and normalaize their height and wight. About 10 % of them continue to remain short (<-2.0SD) and stunting (<-2.0 SD).

Response: We agree with the reviewer that though only 10% of the deliveries with shorter gestational duration likely to be stunted, it is useful to describe. However, there are only a few cases with shorter gestational duration. For instance, five months’ gestational duration, there are only 24 cases and indicated 16 of these cases with stunted children (66.7%)! Similarly, for six months’ gestational duration, there are 62 cases and 34 cases with stunted children (55%). However, we have added a paragraph towards end of the discussion section.

Comment 4: The authors should shorten background. The description about MDGs is too long.

Response: As per the comment, the description about MDGs is shortened.

Comment 5:  Methods and materials 2.2. Predictors) Hence, in the multivariate analysis, the interaction between low birth weight and body mass index is included. (2. Methods and materials 2.3. Statistical analysis)

In order to assess its risk with stunting of children, for important confounders such as BMI, ANC and the interaction terms between them were adjusted in addition to other background characteristics.

Is “BMI of the mothers “ correct?

Response: We checked the computation, BMI of mothers is correct.

Round 2

Reviewer 3 Report

The manuscript has been much improved and is in a nice condition now.